# Physical Activity, Sedentary Behaviour, Weight Status, and Body Composition among South African Primary Schoolchildren

**DOI:** 10.3390/ijerph191811836

**Published:** 2022-09-19

**Authors:** Markus Gerber, Christin Lang, Johanna Beckmann, Rosa du Randt, Kurt Z. Long, Ivan Müller, Madeleine Nienaber, Nicole Probst-Hensch, Peter Steinmann, Uwe Pühse, Jürg Utzinger, Siphesihle Nqweniso, Cheryl Walter

**Affiliations:** 1Department of Sport, Exercise and Health, University of Basel, Grosse Allee 6, CH-4052 Basel, Switzerland; 2Human Movement Science, Nelson Mandela University, P.O. Box 7700, Gqeberha 6031, South Africa; 3Swiss Tropical and Public Health Institute, Kreuzstrasse 2, CH-4123 Allschwil, Switzerland; 4University of Basel, Petersplatz 1, CH-4001 Basel, Switzerland

**Keywords:** accelerometer, body fat, body water, bone mass, fat-free mass, muscle mass, physical activity, schoolchildren, sedentary behaviour, South Africa

## Abstract

**Background:** Over the past decades, childhood overweight has increased in many African countries. We examined the relationship between sedentary behaviour, moderate-to-vigorous physical activity (MVPA), and body composition in South African primary schoolchildren living in peri-urban settings. **Methods:** MVPA was measured via 7-day accelerometry and body composition via bioelectrical impedance analysis in 1090 learners (49.2% girls, M_age_ = 8.3 ± 1.4 years). The relationships between MVPA and sedentary behaviour with the various body composition indicators (body fat and fat-free mass [total, truncal, arms, and legs], bone mass, muscle mass, and body water) were tested with mixed linear regressions. **Results:** The prevalence of overweight and obesity was 9.8% and 6.6%, respectively; 77.1% of the children engaged in ≥60 min of MVPA/day. Girls were more likely to be overweight/obese, to accumulate less than 60 min of MVPA/day, and had significantly higher relative body fat than boys (*p*s < 0.001). Lower MVPA was associated with a higher likelihood of being overweight/obese, higher relative body fat, and lower relative fat-free mass, bone mass, muscle mass, and body water (*p*s < 0.001). For lower sedentary behaviour, the associations with body composition pointed in the opposite direction. **Conclusions:** In this South African setting, girls are a particularly relevant target group for future physical activity interventions to prevent overweight/obesity-related non-communicable diseases in later life.

## 1. Introduction

For the development of overweight and obesity, four critical stages or sensitive periods have been identified: intrauterine life, infancy, the periods of adipose rebound (5–7 years), and adolescence [1]. Evidence highlights that children who become overweight/obese during one of these sensitive periods are at an increased risk of overweight-related non-communicable diseases (NCDs) later in life, such as hypertension [2]. Worldwide, there are serious concerns regarding the increase in overweight and obesity of children and adolescents, especially in urbanized populations and in high-income countries (HICs) [3]. Starting at pre-school age, the prevalence of overweight and obese children has increased steadily during the past 20 years [4,5,6,7,8].

Of note, childhood overweight and obesity has also increased in low- and middle-income countries (LMICs), mainly governed by rapid lifestyle and nutrition transitions [4,5,9,10,11]. In 2011, approximately three out of four overweight/obese children aged below 5 years lived in LMICs [12]. The estimated childhood prevalence of overweight and obesity in Africa was 8.5% in 2010 and 12.7% in 2020 [4]. Compared to a prevalence of only 4.0% in 1990, these are significant increases [13]. In line with these findings, the prevalence of overweight/obesity is rapidly growing among South African school-aged children [14]. Indeed, in the 2013 South African National Health and Nutrition Examination Survey (SANHANNES-1), 7.2% of the 6- to 9-year-old children were classified as overweight/obese, whereas this rate was 10.2% among the 10- to 14-year-old peers. Overweight/obesity was more prevalent among girls than boys and children living in low- compared to high-income settings [15]. Similarly, in a recent study with more than 10,000 South African primary schoolchildren (aged 6–13 years), 8.1% of the children were underweight, whereas 15.4% were overweight or obese [16]. A cross-sectional study of 1136 children (aged 9–13 years) living in urban areas of the Pretoria central region further revealed that girls had a significantly higher percentage of body fat (assessed via thickness of skinfolds) compared to boys, whereas the percentage of body fat was higher among black African compared to white children [17]. Research published over the past five years (2017–2022) confirms that the prevalence of overweight/obesity generally exceeds 15% in South African children and adolescents [18,19,20], an observation which was also made in the Eastern Cape province [21,22]. In one study with 16-year-old adolescents from Limpopo province, the prevalence was in excess of 35% [23].

The trend towards increased body size among South African children has been largely attributed to the process of urbanization and socioeconomic change [24]. Despite the fact that traditional lifestyles prevail in many parts of Africa [13], researchers have observed a transition to Western-oriented lifestyles, particularly in urban settings [25]. The typical Western diet is characterized by energy-dense foods rich in fat and refined sugar, and thus often results in a positive energy balance, especially when physical activity levels are low [13].

Scholars have further argued that if societies fail to implement population-based interventions to reduce overweight/obesity, the steady rise in life expectancy may soon come to an end in modern societies. As a result, children and adolescents of today may live less healthily and potentially die earlier than their parents [26]. To prevent the anticipated trends in overweight/obesity among children, a number of researchers have suggested a focus on key (modifiable) health behaviours [27]. A (positive) energy imbalance due to higher energy intake compared to expenditure is the main physiological cause of excess fat mass accumulation [28]. Accordingly, one possibility to counteract the trend towards increasing childhood overweight/obesity is to ensure that children engage in sufficient daily physical activity. The World Health Organization (WHO) recommends that children and adolescents should accumulate at least 60 min of moderate-to-vigorous physical activity (MVPA) per day.

Studies have shown that physical activity is indeed negatively associated with overweight/obesity [29]. Among young people, an increase in daily MVPA by 10 min was associated with positive effects on some metabolic outcomes such as lower waist circumference and lower fasting glucose [30]. Moreover, negative cross-sectional associations have been reported between physical activity and fat mass in children [31]. However, most of the existing research is based on children living in HICs [32,33], whereas studies on the relationship between physical activity and overweight/obesity and body fat in children living in sub-Saharan Africa are still rare [11,34]. Insufficient physical activity has been found to be an important risk factor for overweight/obesity among adult populations in studies carried out in South Africa [35]. In a study conducted in Johannesburg with 218 children aged 5–10 years, no significant associations were found between moderate physical activity (MPA), vigorous physical activity (VPA), MVPA, and sedentary time and body mass index (BMI)-for-age z-scores (BAZ), weight-for-age (WAZ), and being overweight or obese [36]. By contrast, studies showed that children with poor gross motor skills were more likely to be overweight or obese [37].

The three main objectives of the current study were as follows. First, to examine the prevalence of learners classified as underweight, normal weight, overweight, and obese, to identify the proportion of children with high level of body fat, and to compare whether the prevalence rates were associated with children’s sex and age. Our study adds to the current state of research because there are only few studies on percentage body fat of South African children, in which a device-based approach was used to assess body fat [17]. Second, to examine the linear relationship between sedentary behaviour, MVPA, and several indicators of children’s body composition. Our study will provide insights because in clinical practice, different measures such as BMI or body fat percentage have been used to evaluate the human health risks associated with overweight and obesity, and it is not fully clear which markers are better predictors of future cardiovascular disease [38]. Moreover, data on body composition (going beyond body fat) among South African children is scarce. Such evidence is critical because physical activity not only prevents against high body fat levels, but is also associated with higher muscle mass [39] and favourable bone health [40]. Third, to explore whether children who meet vs. those who do not meet current MVPA recommendations of the WHO (≥60 min of MVPA per day) differ with regard to their weight status and the various indicators of body composition, and whether this relationship is moderated by their sex. These insights are valuable because few studies have simultaneously used objective measures to assess physical activity and body composition in South African children. This is important, as subjective measures are susceptible to bias associated with recall or social desirability, are difficult to complete for young children with limited reading skills and who have a limited cognitive capacity for self-reflection.

## 2. Materials and Methods

### 2.1. Design

Our analyses are based on cross-sectional baseline data stemming from a cluster-randomized controlled trial (https://www.kaziafya.org, accessed on 12 September 2022).

### 2.2. Setting

The trial took place in the Gqeberha (formerly known as Port Elizabeth) region in South Africa, where baseline data were collected in four public schools (41 classes) between February and April 2019 (for specific information, see study protocol) [41]. All schools (quintile 3 schools) were located in peri-urban, marginalized communities. Results from this study have been published previously [42,43,44,45,46,47,48]. However, the objectives articulated above have yet to be examined, and hence, the findings presented in the current publication are original.

### 2.3. Participants and Procedures

Permission for the project was requested from school authorities after they were fully informed and before contact was made with school principals. For all children, written informed consent of a parent/legal guardian was required to participate in the study. Eligibility criteria for schools were: (i) public schools located in marginalized communities; (ii) available facilities to implement physical education lessons; and (iii) currently not engaging in any other research project, or clinical trial. Children were included in the study if they (i) attended grades 1–4; (ii) were not older than 12 years; (iii) had written informed consent from a parent/guardian; and (iv) were not suffering from clinical conditions that prevent participation in physical activity, as determined by qualified medical personnel. At baseline, written informed consent was available for 1369 children (48% girls, 52% boys).

### 2.4. Ethical Considerations

The study was approved by the following institutions: (i) research ethics committee of the Nelson Mandela University (reference number: H18-HEA-HMS-006); (ii) Department of Education of the Eastern Cape Province; and (iii) ‘Ethikkommission Nordwest- und Zentralschweiz’ (EKNZ; reference number: Req-2018-00608). Children identified with severe medical conditions and/or children who were malnourished (based on national guidelines) were referred to local clinics. All procedures were in line with the ethical principles of the Declaration of Helsinki.

### 2.5. Measures

Triaxial accelerometer devices (ActiGraph^®^ wGT3X-BT; Pensacola, FL, USA) were employed to objectively assess children’s physical activity levels. The accelerometers were worn around the hip for seven consecutive days, except for activities involving water contact. A 30 Hz sampling rate was used. Data were recorded continuously. Since the study focused on daytime physical activity, a filter was set to extract only data assessed between 06:00 and 24:00 hours. All recordings were saved in GTX-format. ActiLife software (version 6.13.2; Pensacola, FL, USA) was then applied to analyse the data. We used the Troiano et al. algorithm (default setting) to estimate non-wear time [49]. For the purpose of the present study, only those children who had valid data on ≥4 weekdays and ≥1 weekend day were included [50]. Data were considered valid if wear time was at least 8 h per day [51]. Finally, child-specific cut-points were used to calculate indices of sedentary behaviour and MVPA [52].

Bioelectrical impedance analysis (BIA) was used to assess body composition using a wireless body composition monitor (Tanita MC-580; Tanita Corp., Tokyo, Japan). When BIA was carried out, children wore light clothing, stood barefoot on the metal plates of the machine, and were guided by a research assistant to ensure optimal contact with the device. The same device was used to measure body weight, to the nearest 0.1 kg. A stadiometer was used to assess body height (to the nearest 0.1 cm). To this end, the children stood with their back erect and shoulders against the stadiometer. The WHO growth reference data were used to compute sex-specific BMI z-scores [53]. Children were classified as underweight if they were below the 5th percentile, normal weight if between the 5th percentile up to the 84th percentile, overweight if they were between the 85th and the 94th percentile, and as obese if their BMI z-score was equal to or greater than the 95th percentile. Children with body fat percentages ≥30% were classified as having high body fat [54].

### 2.6. Statistical Analyses

We first calculated descriptive statistics for each of the assessed variables. The Kolmogorov-Smirnov test was used to test normality. We carried out mixed multinomial regression analyses to examine the association between MVPA and sedentary behaviour with children’s weight status (underweight, normal weight, overweight, and obese). Mixed logistic regressions were used to test the association between MVPA and high body fat (≥30%) levels. Additionally, we used mixed linear regressions to examine the relationship between MVPA and sedentary behaviour with the various body composition indicators (body fat and fat-free mass [total, truncal, arms, and legs], bone mass, muscle mass, and body water. For both mixed multinomial, logistic, and linear regression analyses, school class was used as a random intercept to account for the nested nature of the data (learners nested in classes). All regression analyses were controlled for participants’ sex, age, and accelerometer wear time. In addition, to find out whether children who accomplished current MVPA recommendations set forth by WHO (i.e., ≥60 min of MVPA per day) differ from their less active peers with regard to their weight status and the various indicators of body composition, we carried out χ^2^-tests and a series of two-way analyses of covariance (ANCOVAs), with the following factors: (a) physical activity (meeting vs. not meeting MVPA recommendations), sex (girls vs. boys), and the interaction between sex and physical activity. All analyses were controlled for participants’ age and accelerometer wear-time. The effect sizes of ANCOVAs were interpreted as follows: η^2^ < 0.01: negligible, η^2^ ≥ 0.01: small, η^2^ ≥ 0.06: medium, and η^2^ ≥ 0.138: large. All statistical analyses were carried out with SPSS 26 for Mac (IBM Corporation; Armonk, NY, USA), and statistical significance was set at *p* < 0.05.

## 3. Results

### 3.1. Sample Characteristics and Between-Sex Differences

The sample consisted of 1090 learners (553 boys, 537 girls) who had complete data across all study variables (Table 1). The mean age was *M* = 8.3 ± 1.4 years. The majority of the children self-identified them as black African (*n* = 565, 51.8%) or coloured (*n* = 509, 46.7%). The average height of the learners was *M* = 124.6 ± 9.1 cm, and the average weight *M* = 25.3 ± 6.7 kg. The mean BMI was *M* = 16.08 ± 2.60 kg/m^2^. Regarding weight status, 49 learners (4.5%, 95% confidence interval (CI): 3.3–5.9%) were classified as underweight, 862 as normal weight (79.1%, 95% CI: 76.5–81.5%), 107 as overweight (9.8%, 95% CI: 8.1–11.7%), and 72 as obese (6.6%, 95% CI: 5.2–8.2%). Girls were over-represented among underweight (girls: 7.1%; boys: 2.0%), overweight (girls: 11.4%; boys: 8.3%), and obese children (girls: 9.3%; boys: 4.0%), whereas boys were over-represented among normal weight children (girls: 72.3%; boys: 85.7%, χ^2^(1, 1090) = 36.22, *p* < 0.001). The mean percentage body fat was *M* = 22.6 ± 5.3%. In total, 84 learners were classified as having high body fat levels (7.7%, 95% CI: 6.2–9.5%), with this percentage being higher among girls (11.5%) than boys (4.0%), χ^2^(1, 1090) = 21.94, *p* < 0.001. On average, the children engaged in *M* = 609.0 ± 69.1 min/day of sedentary behaviour and in *M* = 82.2 ± 27.7 min/day of MVPA. In total, 840 learners met current MVPA recommendations (77.1%), with a considerably higher proportion of boys (91.1%) being sufficiently active than girls: (62.6%, χ^2^(1, 1090) = 125.91, *p* < 0.001). For a complete overview of descriptive statistics, see Table 1.

Kolmogorov–Smirnov tests showed that with the exception of sedentary behaviour, none of the metric variables were normally distributed. Because skewness and kurtosis values were below |2| and/or ≥|7|, respectively, no evidence for severe non-normality was identified [55]. Thus, none of the variables had to be log-transformed before calculating the inferential statistics.

### 3.2. Relationship between Sedentary Behaviour, MVPA, Weight Status, and Body Composition

The multinomial regression analyses showed that increased sedentary behaviour increased the likelihood of learners to be classified as obese (in reference to normal weight) (Table 2). By contrast, higher levels of MVPA were associated with a decreased risk of being classified as overweight or obese. Neither sedentary behaviour nor MVPA were associated with an altered risk of being classified as underweight. Mixed logistic regression analyses further showed that increased sedentary behaviour was associated with an increased risk of having high body fat. The opposite was the case for learners with higher MVPA levels.

A similar picture emerged for the other body composition indicators. Thus, while more sedentary behaviour was associated with higher percentage body fat (in the whole body and specific body-parts), negative relationships were found in the mixed linear regression analyses between sedentary behaviour and fat-free mass, bone mass, muscle mass, and body water. By contrast, higher levels of MVPA were associated with lower body fat, higher fat-free mass, higher bone mass, higher muscle mass, and higher body water. Figure 1 displays scatterplots (including a regression line), separately for sedentary behaviour and MVPA, and for all body composition indicators.

### 3.3. Differences between Children Who Meet vs. Do Not Meet Recommended MVPA Levels

The weight status categories are distributed differently between learners who meet vs. do not meet current MVPA recommendations, both in boys and girls (Figure 2A). In boys, children who accomplish MVPA standards are less likely to be underweight, overweight, and obese. In girls, children with sufficiently high MVPA levels are less likely to be overweight and obese; however, they are also slightly overrepresented in the underweight category. χ^2^-tests confirm that the differences are statistically significant in the total sample, χ^2^(3, 1090) = 73.49, *p* < 0.001, in girls, χ^2^(3, 537) = 36.98, *p* < 0.001, and in boys, χ^2^(3, 553) = 9.62 *p* < 0.01.

Independent of the sex of the children, learners with sufficiently high MVPA levels were less likely to be classified in the group with high body fat levels (Figure 2B). The results of the χ^2^-tests were statistically significant for the total sample, χ^2^(3, 1090) = 61.24, *p* < 0.001, for girls, χ^2^(3, 537) = 33.44, *p* < 0.001, and for boys, χ^2^(3, 553) = 8.36, *p* < 0.05.

Boys and girls differed with regard to all indicators of body composition (Table 3). Thus, after adjusting for age, girls had higher percentage of body fat, lower percentage of fat-free mass, lower percentage of bone mass, lower percentage of muscle mass, and lower percentage of body water (all *p* < 0.001). Children’s sex explained between 3.2% and 26.0% of variance in the various body composition indicators. In addition, learners with sufficient MVPA levels differed from their less active peers across all outcome variables. Thus, more physically active peers had a lower percentage of body fat. By contrast, they had higher relative fat-free mass, bone mass, muscle mass, and body water (all *p* < 0.001). For the factor MVPA, the level of explained variance ranged between 2.0% and 2.9%. Finally, the fact that none of the two-way interactions was statistically significant highlights that the association between MVPA and body composition was not moderated by the sex of the learners.

## 4. Discussion

The present study is among the first to determine how objectively assessed physical activity is associated with weight status and body composition, assessed via BIA technology among South African primary schoolchildren. The key findings are that higher levels of sedentary behaviour are associated with a higher likelihood of being classified as obese, with higher relative body fat, as well as lower relative fat-free mass, bone mass, muscle mass, and body water. The contrary was the case for higher engagement in MVPA. In line with these findings, children who met current WHO recommendations for physical activity (≥60 min of daily MVPA) were less likely to be classified as overweight or obese, had lower relative body fat, as well as higher relative fat-free mass, bone mass, muscle mass, and body water. Significant differences between boys and girls were found across all body composition indicators. However, the association between MVPA and body composition was not moderated by children’s sex.

Previous research suggested that the prevalence of overweight and obesity is rapidly increasing among school-aged children in sub-Sahara Africa [14]. Whereas in our study, 16.4% of the children were classified as overweight or obese, the prevalence was markedly lower in the nationally representative SANHANNES-1 study (published in 2013), in which 7.2% of the 6- to 9-year-old children and 10.2% of children aged 10–14 years were classified as overweight/obese [15]. The higher numbers can be attributed to the fact that overweight/obesity rates have steadily increased over the past several years but might also be reflective of the fact that our study took place in relatively poor, marginalized urban communities. Our study further suggests that in urban settings, overweight/obesity is meanwhile 3.6 times more prevalent than underweight in a sample of predominantly black African and coloured children. This is in line with the Birth-to-20 Plus study, where the prevalence of underweight among 5-year-old black African was 3%, whereas the prevalence of overweight was 11% [56]. We also found that already among this young age group, a considerable number of learners (7.7%) had high body fat levels. Moreover, our study confirmed that girls have a higher risk of being classified as overweight and obese, compared to boys [15].

Our results are critical for three major reasons. First, overweight and obesity seem to be relatively stable and track from childhood into adolescence and adulthood [57]. More specifically, in a previous study, obese girls aged 4–8 years had a 42 times higher odds of being obese at the age of 16–18 years, whereas in boys the odds was almost 20 times higher [58]. Second, overweight/obese children have a significantly higher clustered cardiovascular risk [59]. Third, childhood overweight and obesity are strong predictors of type 2 diabetes mellitus, hypertension, and dyslipidaemia during adulthood [60], which in turn are associated with an increased cardiovascular disease risk and premature death [61].

Our study further confirms previous research with South African primary schoolchildren in which a significant relationship was identified between children’s time spent in sedentary behaviour and body mass. For instance, it could be shown that spending more than 4 h per day in front of a screen doubled the risk of being overweight [62]. Total body fat and intra-abdominal adipose tissue constitute the most important clinical indicators of body fat, as they are most closely related with hypertension, hyper-insulinemia, type 2 diabetes, and dyslipidaemia [1]. The data of our study confirm that higher levels of sedentary behaviour are associated with higher relative body fat in different parts of the body (arms, legs, and trunk), whereas the opposite was the case for higher MVPA engagement.

Although equivocal results have been reported in previous research [28,31,36,37], our study corroborates that engaging in at least 60 min of MVPA per day has the potential to prevent overweight/obesity among South African schoolchildren. We acknowledge that the cross-sectional nature of our data prevents us from drawing conclusions about cause and effect. Nevertheless, it is important to note that the potential to prevent the development of overweight/obesity in initially non-overweight children has previously been demonstrated in a study with (older) South African primary schoolchildren [63]. A particular strength of our study was that both physical activity and body composition were assessed with a device-based approach. Moreover, our study also explored whether current MVPA recommendations can be used as a meaningful benchmark to make recommendations to stakeholders and parents. Based on the present analyses, we can conclude that 60 min of daily MVPA is indeed a relevant anchorage point that equally applies for boys and girls. This is noteworthy as between-sex differences in overweight/obesity have been ascribed to higher MVPA levels among boys and lower sedentary time compared to girls [17].

The significant relationships between MVPA and body composition can be attributed to several mechanisms and can be regarded as a two-way process [57]. On one hand, low levels of physical activity may contribute to the accumulation of fat mass, whereas higher levels of physical activity may increase lean mass [64]. On the other hand, body composition may affect physical activity; for example, higher levels of adiposity may impede exercise directly and indirectly [65], such as obese children who have lower perceived and/or actual motor performance [66], thus are less likely to exercise [67].

In line with previous research, significant between-sex differences in body composition were found in our study [68]. As highlighted by WHO, females are at higher risk to become overweight/obese than males [69]. In South Africa, this seems to apply particularly to black African women who have especially high risks of being overweight/obese or having abdominal obesity [70]. Given the young age of our study participants, we did not expect large between-sex differences. Typically, as a consequence of maturation, differences in body composition increase between boys and girls during adolescence. More specifically, from the age of 11 to 16 years, fat-free mass increases by about 40% in girls and 90% in boys [71]. Nevertheless, between-sex differences were of substantial magnitude in our sample of grade 1–4 learners. This might be due to the fact that in the African culture, increased body fatness is regarded as a sign of health and wealth [13], which might motivate parents to over-feed their children, hereby increasing their risk of becoming overweight/obese [72].

Our study has several strengths such as the objective assessment of both dependent and independent variables, the relatively large sample size, and the inclusion of school class as a random intercept in the regression analyses. It is also noteworthy that BIA allowed us to make a distinction between fat and fat-free mass (including bone mass and muscle mass), as they are associated with physical activity in opposite directions [57].

Our study also has some limitations that are offered for discussion. First, as highlighted previously, given the cross-sectional nature of the study, no causal interpretation of the data is permitted. Second, whereas prior research demonstrated that BIA provides a valid estimate of overall body fat [73], further evidence is needed to show that body fat assessed via BIA and dual-energy X-ray absorptiometry (DXA) are sufficiently correlated in specific body segments. While we are not aware of such validation studies among children, research with young adults showed that DXA estimations were highly correlated with Tanita-based BIA (total body: *r* = 0.94, trunk: *r* = 0.82, legs: *r* = 0.90, and arms: *r* = 0.83; all *p* < 0.001) [74]. Third, our sample focused on marginalized, peri-urban communities in South Africa. Thus, no generalization is possible beyond this subgroup, and further research is warranted with children from different socioeconomic backgrounds or children who live in rural areas or other regions of South Africa. Fourth, while device-based assessments of physical activity allow a distinction between activities at different intensity levels, they do not provide information about the type of activity and the setting, in which this activity takes place. Thus, we cannot conclude whether higher MVPA levels are due to greater distance to school, household chores, or informal or structured exercise/sport activities. Furthermore, it is a general weakness of accelerometry that water sports cannot be accurately assessed. However, all children from our study lived in marginalized areas, which were located far away from the sea and there are no public swimming pools. Hence, water sports in our sample are highly unlikely. Moreover, although no “gold” standard method exists for the assessment of physical activity in epidemiological research, accelerometry is often seen as the most objective technique to measure gross body movement [75]. Nevertheless, accelerometry is not without disadvantages and also requires subjective decisions by the researchers (e.g., with regard to algorithm to estimate wear time, minimal wear time per day, number of minimally required valid days, and selection of cut-offs for different intensity types) [75,76]. The fact that many children might not wear the device across the entire day or forget to put on the device on some days might negatively impact the precision of the estimates. This is why researchers generally set minimal standards. However, wear time can vary considerably even between children who fulfill minimal standards. We addressed this issue by controlling regression analyses for accelerometer wear time to ensure that relationships are indeed attributable to differences in MVPA and not a mere reflection of different accelerometer wear times. Despite these potential shortcomings, a recent meta-analysis showed that accelerometry provides accurate estimates of MVPA in children [77]. Finally, in this article, we focused on physical activity levels as a proxy for energy expenditure, which is only one component in the energy balance. Thus, future studies should assess more specific information about energy intake in- and outside the school context to obtain a more complete picture about what can be achieved via the promotion of physically active lifestyles.

## 5. Conclusions

This is one of the first studies that used accelerometers and bioelectrical impedance to examine the association between physical activity and body composition in a sample of South African primary schoolchildren. Our findings expand the existing literature in that regular physical activity and lower levels of sedentary behaviour are associated with more favourable body composition and a lower odds of being classified as overweight/obese. The majority of learners engaged in sufficient level of MVPA; yet, the percentage of insufficiently active children was substantially higher among girls. Additionally, girls were more likely to be classified as overweight/obese or having high levels of body fat. Accordingly, girls are a particularly relevant target group for future interventions to prevent overweight/obesity-related NCDs in later life. Given that school is the place where children spend a considerable proportion of their waking hours, schools have the potential to play a crucial role in the promotion of physical activity and the adoption of a healthy diet.

## Figures and Tables

**Figure 1 ijerph-19-11836-f001:**
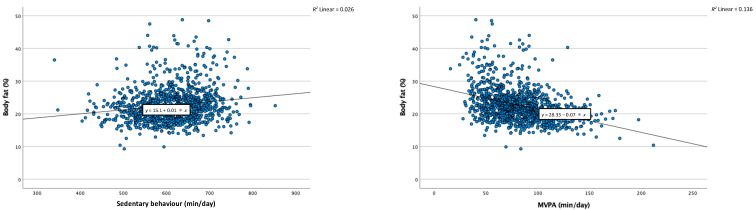
Scatterplots with regression line, representing the relationships between sedentary behaviour and moderate-to-vigorous physical activity (MVPA) with the various body composition indicators. Notes. min = Minutes. *R*^2^ = Total explained variance by regression model.

**Figure 2 ijerph-19-11836-f002:**
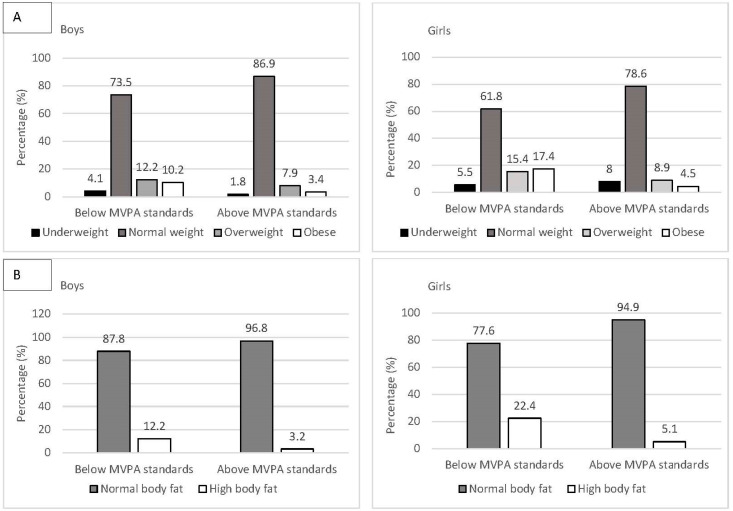
Percentage of students in each (**A**) weight status group (underweight, normal weight, overweight, and obese) and (**B**) with normal vs. high body fat levels, separately for those who met vs. do not meet current moderate-to-vigorous physical activity (MVPA) standards (≥60 min of MVPA/day) and separately for boys and girls.

**Table 1 ijerph-19-11836-t001:** Descriptive statistics for the total sample for physical activity, sedentary behaviour, body composition measures, and anthropometric variables.

Metric Variable	*M*	*SD*	Min	Max	Skew	Kurt
Accelerometry						
Sedentary behaviour (min/day)	609.0	69.1	339.8	852.3	−0.17	0.22
MVPA (min/day)	82.2	27.7	15.7	211.9	0.61	0.58
Body composition: overall						
Body fat (in kg)	5.9	3.1	1.7	25.1	3.00	11.42
Body fat (in %)	22.6	5.3	9.3	48.8	1.45	3.38
Fat-free mass (in kg)	19.4	4.2	10.6	38.1	0.92	1.31
Fat-free mass (in %)	77.4	5.6	51.1	90.6	−1.45	3.36
Bone mass (in kg)	1.1	0.2	0.6	2.0	0.69	0.90
Bone mass (in %)	4.4	0.5	2.7	5.6	−0.30	0.16
Muscle mass (in kg)	18.3	4.0	10.0	36.2	0.93	1.33
Muscle mass (in %)	73.1	4.9	48.7	85.3	−1.48	3.54
Body water (in kg)	14.2	3.1	7.8	27.9	0.92	1.30
Body water (in %)	56.7	3.9	37.5	68.4	−1.40	3.35
Body composition: arms						
Body fat (in kg)	0.7	0.4	0.2	3.9	3.23	14.33
Body fat (in %)	33.5	5.7	16.7	61.9	0.91	2.07
Fat-free mass (in kg)	1.4	0.5	0.3	3.5	0.93	1.37
Fat-free mass (in %)	66.5	5.7	38.1	83.3	−0.91	2.07
Body composition: legs						
Body fat (in kg)	2.4	1.3	0.4	10.8	2.93	11.38
Body fat (in %)	5.2	0.8	9.8	51.8	0.76	1.88
Fat-free mass (in kg)	5.6	1.8	1.6	14.5	1.10	2.23
Fat-free mass (in %)	70.7	5.2	48.2	90.2	−0.76	1.88
Body composition: trunk						
Body fat (in kg)	2.8	1.5	0.3	11.4	2.81	10.19
Body fat (in %)	17.8	5.3	3.0	44.1	1.48	3.61
Fat-free mass (in kg)	12.4	2.0	7.6	21.1	0.71	0.82
Fat-free mass (in %)	82.2	5.3	55.8	97.2	−1.47	3.61
Age and anthropometric measures						
Age (in years)	8.3	1.4	5.7	13.2	0.28	−0.72
Height (in cm)	124.6	9.1	102.0	152.0	0.21	−0.34
Weight (in kg)	25.3	6.7	13.5	61.2	1.74	4.64
BMI (in kg/m^2^)	16.1	2.6	10.5	29.9	2.11	6.14
Categorical variables	*N*	%				
Sex						
Boys	553	50.7				
Girls	537	49.3				
Weight status						
Underweight	49	4.5				
Normal weight	862	79.1				
Overweight	107	9.8				
Obese	72	6.6				
High body fat						
No	1006	92.3				
Yes	84	7.7				

Notes. *M* = Mean. *SD* = Standard deviation. Min = Minimum. Max = Maximum. Skew = Skewness. Kurt = Kurtosis. *N* = Number of students. min = Minutes. kg = Kilograms. cm = Centimeters. m = Meters. MVPA = Moderate-to-vigorous physical activity. BMI = Body mass index.

**Table 2 ijerph-19-11836-t002:** Results of mixed multinomial, logistic, and linear regression analyses to predict body composition via sedentary behaviour and moderate-to-vigorous physical activity.

	Sedentary Behaviour			MVPA		
Mixed Logistic Regression	*B*	*SE*	95% CI	*B*	*SE*	95% CI
Weight status						
Normal weight (reference)	0			0		
Underweight	0.005	0.003	−0.001; 0.010	0.006	0.006	−0.005; 0.019
Overweight	0.001	0.002	−0.002; 0.005	−0.016 **	0.005	−0.025; −0.006
Obesity	0.005 *	0.002	0.001; 0.010	−0.037 ***	0.007	−0.051; −0.024
High body fat ^a^						
No (reference)	0			0		
Yes	0.006 **	0.002	0.002; 0.010	−0.040 ***	0.007	−0.054; −0.027
**Mixed linear regression**						
Body composition: overall						
Body fat (in %)	0.009 ***	0.003	0.004; 0.015	−0.048 ***	0.006	−0.060; −0.037
Fat-free mass (in %)	−0.010 ***	0.003	−0.015; −0.004	0.049 ***	0.006	0.037; 0.060
Bone mass (in %)	−0.001 **	0.000	−0.001; 0.000	0.003 ***	0.001	0.002; 0.004
Muscle mass (in %)	−0.009 ***	0.003	−0.014; −0.004	0.046 ***	0.006	0.035; 0.056
Body water (in %)	−0.007 ***	0.002	−0.011; −0.003	0.036 ***	0.004	0.027; 0.044
Body fat in specific body parts						
Body fat in arms (in %)	0.014 ***	0.003	0.008; 0.020	−0.055 ***	0.006	−0.068; −0.043
Body fat in legs (in %)	0.007 **	0.003	0.002; 0.012	−0.040 ***	0.005	−0.051; −0.029
Body fat in the trunk (in %)	0.011 ***	0.003	0.006; 0.017	−0.052 ***	0.006	−0.064; −0.040

Notes. *B* = Unstandardized beta coefficient. *SE* = Standard error. CI = Confidence interval. MVPA = Moderate-to-vigorous physical activity. ^a^ High body fat is defined as percentage body fat ≥ 30%. All analyses controlled for age, sex, and accelerometer wear time. School class was used as a random intercept across all analyses. * *p* < 0.05. ** *p* < 0.01. *** *p* < 0.001.

**Table 3 ijerph-19-11836-t003:** Differences in body composition between students who meet vs. do not meet current moderate-to-vigorous physical activity standards (≥60 min of MVPA/day), with sex as a moderator.

	Not Meeting MVPA Standards (*n* = 250)				Meeting MVPA Standards (*n* = 840)				Main Effect: MVPA		Main Effect: Sex		Interaction Effect: MVPA by Sex	
	Girls (*n* = 201)		Boys (*n* = 49)		Girls (*n* = 336)		Boys (*n* = 504)							
	*M*	*SD*	*M*	*SD*	*M*	*SD*	*M*	*SD*	*F*	η^2^	*F*	η^2^	*F*	η^2^
Body composition: overall														
Body fat (in %)	26.4	6.3	22.1	6.4	23.4	3.9	20.6	4.4	30.31 ***	0.027	68.96 ***	0.060	2.59	0.002
Fat-free mass (in %)	73.7	6.3	77.9	6.4	76.6	3.9	79.5	4.5	30.37 ***	0.027	68.43 ***	0.059	2.43	0.002
Bone mass (in %)	3.9	0.4	4.6	0.5	4.1	0.3	4.7	0.4	22.20 ***	0.020	377.61 ***	0.260	1.08	0.001
Muscle mass (in %)	69.7	6.0	73.4	5.9	72.5	3.7	74.8	4.2	29.88 ***	0.027	54.46 ***	0.048	2.71	0.002
Body water (in %)	53.9	4.6	57.1	4.8	56.1	2.9	58.2	3.3	28.45 ***	0.026	70.90 ***	0.061	3.05	0.003
Body fat in specific body parts														
Body fat in arms (in %)	37.3	6.6	32.9	6.3	34.3	4.6	31.5	5.0	25.89 ***	0.023	59.20 ***	0.052	3.35	0.003
Body fat in legs (in %)	33.1	5.3	28.0	6.1	30.8	3.4	26.9	4.7	23.02 ***	0.021	122.94 ***	0.102	2.23	0.002
Body fat in trunk (in %)	21.2	6.7	17.9	6.4	18.1	4.2	16.1	4.5	32.08 ***	0.029	36.12 ***	0.032	2.37	0.002

Notes. *n* = Number of participants. *M* = Mean. *SD* = Standard deviation. *F* = *F*-value in analysis of covariance. η^2^ = Eta squared value. MVPA = Moderate-to-vigorous physical activity. All analyses controlled for age and accelerometer wear time. *** *p* < 0.001.

## Data Availability

After acceptance of this article, data used for data analysis will be made publicly available as online Appendix A.

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
