# Peer review of "Physical Activity, Sedentary Behaviour, Weight Status, and Body Composition among South African Primary Schoolchildren"

_ijerph, 2022, doi:10.3390/ijerph191811836_

Round 1

Reviewer 1 Report

Brief summary

Three main goals:

1.       To examine the prevalence of learners classified as underweight, normal weight, overweight and obese, to identify the proportion of children with high level of body fat, and to compare whether the prevalence rates are associated with children’s sex and age.

2.       To examine the linear relationship between sedentary activity, MVPA and several indicators of children’s body composition.

3.       To explore whether children who meet versus those who do not meet current MVPA recommendations of the WHO (≥ 60 minutes of MVPA per day) differ with regard to their weight status and the various indicators of body composition, and whether this relationship is moderated by their sex.

Main contributions: The research provides insights about the status of marginalized children in south Africa related to MVPA and its relationship to overweight, obesity, body fat and sex differences.

Strengths: Representative sample, proper design, data collection and data analysis, deep and thorough discussion.

General comments

Is the manuscript clear, relevant for the field and presented in a well-structured manner?

Yes.

Are the cited references mostly recent publications (within the last 5 years) and relevant? Does it include an excessive number of self-citations?

Only one reference from 2020, two references from 2019 and one from 2019.

Is the manuscript scientifically sound and is the experimental design appropriate to test the hypothesis?

Yes

Are the manuscript’s results reproducible based on the details given in the methods section?

Yes

Are the figures/tables/images/schemes appropriate? Do they properly show the data? Are they easy to interpret and understand? Is the data interpreted appropriately and consistently throughout the manuscript? Please include details regarding the statistical analysis or data acquired from specific databases.

Figures and tables are appropriate and easy to understand. The data has been interpreted appropriately.

Are the conclusions consistent with the evidence and arguments presented?

Discussion and conclusions are consistent and thorough, and reflect the results obtained. However, references from the last 2-3 years should be included.

Please evaluate the ethics statements and data availability statements to ensure they are adequate.

Ethics and data availability statements are adequate. Data will be included as supplementary material when published, according to the statement.

Specific comments

1.       Is it possible to find more recent data about obesity? The most recent reference stated in lines 45-56 is from 2017. It would provide more insight to add information about last 2-3 years. Maybe government data? National surveys (e.g. sports habits)? I do not know if those types of surveys are carried out in South Africa.

2.       Sections 3 (Measures) and 4 (Statistical analyses) should be included within section 2 (materials and methods), as a subsection.

3.       Accelerometer: It is not clear for me, reading lines 145-154, how would you assess physical activity levels in children who made water sports (e.g. swimming). If none of the subjects had this issue, it is OK. But I believe it would be better if this situation was specified in the paper, if it happened.

4.       There are few times throughout the paper in which you use the term “multinominal” instead of “multinomial”. Please correct it.

5.       The majority of references are not recent. Only four references from the last 5 years (2017-2022). I believe more recent references related to the field must be included, both in the introduction and the discussion sections. 

Author Response

Reviewer #1:

Brief summary

Three main goals

  1. To examine the prevalence of learners classified as underweight, normal weight, overweight and obese, to identify the proportion of children with high level of body fat, and to compare whether the prevalence rates are associated with children’s sex and age.
  2. To examine the linear relationship between sedentary activity, MVPA and several indicators of children’s body composition.
  3. To explore whether children who meet versus those who do not meet current MVPA recommendations of the WHO (≥ 60 minutes of MVPA per day) differ with regard to their weight status and the various indicators of body composition, and whether this relationship is moderated by their sex.

Main contributions: The research provides insights about the status of marginalized children in south Africa related to MVPA and its relationship to overweight, obesity, body fat and sex differences.

Strengths: Representative sample, proper design, data collection and data analysis, deep and thorough discussion.

Response: We thank Reviewer #1 for the accurate summary of the main goals, contributions and strengths of our study.

General Comments

Is the manuscript clear, relevant for the field and presented in a well-structured manner?

Yes. 

Are the cited references mostly recent publications (within the last 5 years) and relevant? Does it include an excessive number of self-citations?

Only one reference from 2020, two references from 2019 and one from 2019. 

Is the manuscript scientifically sound and is the experimental design appropriate to test the hypothesis?

Yes

Are the manuscript’s results reproducible based on the details given in the methods section?

Yes

Are the figures/tables/images/schemes appropriate? Do they properly show the data? Are they easy to interpret and understand? Is the data interpreted appropriately and consistently throughout the manuscript? Please include details regarding the statistical analysis or data acquired from specific databases.

Figures and tables are appropriate and easy to understand. The data has been interpreted appropriately. 

Are the conclusions consistent with the evidence and arguments presented?

Discussion and conclusions are consistent and thorough, and reflect the results obtained. However, references from the last 2-3 years should be included. 

Please evaluate the ethics statements and data availability statements to ensure they are adequate.

Ethics and data availability statements are adequate. Data will be included as supplementary material when published, according to the statement.

Response: We thank Reviewer #1 for the overall positive appraisal of our research. While revising, we updated the reference list and provide cross-reference to a few more recent publications.

Specific Comments

  1. Is it possible to find more recent data about obesity? The most recent reference stated in lines 45-56 is from 2017. It would provide more insight to add information about last 2-3 years. Maybe government data? National surveys (e.g. sports habits)? I do not know if those types of surveys are carried out in South Africa.

Response: We agree with this criticism. We have included more recent data on obesity and physical activity in African children (see revised manuscript, page 2, lines 15-19).

  1. Sections 3 (Measures) and 4 (Statistical analyses) should be included within section 2 (materials and methods), as a subsection.

Response: We agree, and hence, we have changed the section numbering accordingly.

  1. Accelerometer: It is not clear for me, reading lines 145-154, how would you assess physical activity levels in children who made water sports (e.g. swimming). If none of the subjects had this issue, it is OK. But I believe it would be better if this situation was specified in the paper, if it happened.

Response: Thank you for this comment. Your statement is correct: It is a general weakness of accelerometry that water sports cannot be assessed accurately. However, we would like to emphasize that all the children from our study come from marginalized areas which are relatively far away from the sea and where no public swimming pools are available. We therefore believe that participation in water sports is quite uncommon in our sample, and thus had no effect on our findings. In response to your point, we have addressed this issue in the discussion as a limitation (see revised manuscript, page 12, lines 1-4).

  1. There are few times throughout the paper in which you use the term “multinominal” instead of “multinomial”. Please correct it.

Response: We thank Reviewer #1 for reading our piece so carefully. We have made the necessary corrections.

  1. The majority of references are not recent. Only four references from the last 5 years (2017-2022). I believe more recent references related to the field must be included, both in the introduction and the discussion sections.

Response: We accept this criticism. As stated above, we have updated our list of references, and have included a few more recent papers which were published in the past five years (2017-2022).

Reviewer 2 Report

This research examined the relationship between sedentary activity, moderate-to-vigorous physical activity (MVPA) and body composition in South African primary schoolchildren living in peri-urban settings.

In my opinion, the research is not really original and adequately important that I am unsure if our readers may be interested in reading it, although efforts are seen.

English and the flow of content throughout the manuscript need to be enhanced.

Grammatical and syntax errors seen across the manuscript shall be corrected too.

References are very old with limited in the recent years referred, newer literature is needed in this manuscript.

The introduction does not reflect the importance of the study; in fact, it blurred out the focus on HIC and LMICs, as well as the purposes of conducting this research.

Some of the eligibility criteria are redundant. For example: "(iii) currently not engaging in any other research project, or clinical trial." and "(iv) were not participating in other research project or clinical trial" are the same.

Abbreviations in tables shall be explained.

Author Response

Reviewer #2:

*************************************************************************************

General comments

This research examined the relationship between sedentary activity, moderate-to-vigorous physical activity (MVPA) and body composition in South African primary schoolchildren living in peri-urban settings. In my opinion, the research is not really original and adequately important that I am unsure if our readers may be interested in reading it, although efforts are seen. 

Response: We thank Reviewer #2 for the summary of the main purposes of our study. While we agree that the association between physical activity and body composition has been assessed previously in high income countries (see revised manuscript, page 1, lines 37-40), there is a paucity of studies on the relationship between physical activity and overweight/obesity and body fat in children from sub-Saharan Africa (see revised manuscript, page 1, lines 43-45). Yet, it is critical to have quality data from different African settings, as substantial lifestyle changes currently take place, particularly in urban settings, characterized by a transition towards more Western-oriented lifestyles (see revised manuscript, page 2, lines 20-25). Hence, we believe that this is a timely topic that deserves attention. We would also like to emphasize that few studies have so far used a device-based approach to link physical activity and body fat in South African children. Indeed, only few studies have combined accelerometry and body impedance to “objectively” assess physical activity and body composition (see revised manuscript, page 3, lines 7-15). Finally, we have also highlighted in our piece that data on body composition going beyond body fat percentage is scarce among South African children (see revised manuscript, page 3, lines 4-7). Against this background, we believe that the current paper adds to the existing state of knowledge in different meaningful ways, which justifies publication in an international public health-oriented journal.

English and the flow of content throughout the manuscript need to be enhanced.

Response: While revising our manuscript, we have asked several English native speakers to check the flow and language. Consequently, adjustments were made to further improve flow and language.

Grammatical and syntax errors seen across the manuscript shall be corrected too.

Response: As mentioned in response to the previous point, several English native speakers have carried out a thorough language check. We hope that all issues with regard to grammar and syntax are now resolved.

References are very old with limited in the recent years referred, newer literature is needed in this manuscript.

Response: We accept this criticism, which is in line with feedback provided by Reviewer #1. As a consequence, we have updated our list of references and have included several newer studies that were published in the past 5 years (2017-2022).

The introduction does not reflect the importance of the study; in fact, it blurred out the focus on HIC and LMICs, as well as the purposes of conducting this research. 

Response: As explained above, we believe that we have sufficiently explained how the current study adds to the existing body of literature. Nevertheless, we agree with Reviewer #2’s point of view that we could have highlighted more explicitly that the relationship between physical activity and body composition has, thus far, mainly been assessed in high-income countries. We now emphasize more prominently that our piece makes an important contribution for a marginalized community in sub-Saharan Africa (see revised manuscript, page 2, lines 44-46).

Specific comments

Some of the eligibility criteria are redundant. For example: "(iii) currently not engaging in anyother research project, or clinical trial." and "(iv) were not participating in other research project orclinical trial" are the same.

Response: We thank Reviewer #2 for identifying this redundancy in the eligibility criteria. We have changed the manuscript accordingly (see revised manuscript, page 3, line 44).

Abbreviations in Tables shall be explained.

Response: This issue has been addressed in Tables 1-3 and Figures 1 and 2 Hence, we now explain all abbreviations in the Notes sections.

Reviewer 3 Report

Dear Autours,

I'm happy to review your interesting paper.

Please refer to my comments.

I hope my coments make your paper well.

Thanks,

Author Response

Reviewer #3:

*********************************************************************************

General comments

I'm happy to review your interesting paper. Please refer to my comments. I hope my comments make your paper well. This is an important paper for examining the relationship between sedentary activity, MVPA and body composition in primary schoolchildren. I had some comments that the authors may like to consider.

Response: We thank Reviewer #3 for the positive appraisal of our research. We have carefully considered all comments and have adapted the manuscript accordingly.

Specific comments

Introduction/Background:

Why do you insist on income? (Page 1, Line 35‐37, 40‐41)

Response: Our intention was to emphasize that overweight and excessive body fat is not only an issue in high-income countries, but has become more and more important in low- and middle-income countries, due to rapid transitions in lifestyle and nutrition towards more sedentary lifestyles and more unhealthy/Western diets.

This sentence (Line 40‐41) mentions the reverse of the previous sentence (Line 35‐37). You need to refer to (explain) previous studies accurately.

Response: We are unsure why Reviewer #3 sees a contradiction. To wit: on page 1, lines 35-37, we state that worldwide, there are serious concerns regarding the increase in overweight and obesity in children and adolescents, especially in urbanized populations and high-income countries. On line page 1, lines 42-43, we state that childhood overweight and obesity has also increased in low- and middle-income countries (LMICs) due to lifestyle and nutrition transitions. Hence, no corrective actions have been taken.

You had better describe clearly and definitely. For example, “Among young people, an increase in daily MVPA by 10 minutes was associated with positive effects on some metabolic outcomes such as waist circumference and fasting glucose [17].”  Positive effects on waist circumstance, fasting glucose → Tighten waist, decreasing fasting glucose. (Page 2, Line 77‐80).

Response: We agree that more precision is needed, and hence, we have changed our statement in line with this recommendation (see revised manuscript, page 3, 41-42).

You mention and point out “subjective measures”. I think it is difficult for children to read and describe (fill in) questionnaires regarding what they did accurately (Page 3, Line 111).

Response: This is an important observation. We have added that beyond issues associated with recall or social desirability, subjective measures are difficult to complete for young children who are still learning to read and write and have limited cognitive capacity for self-reflection (see revised manuscript, page 3, 21-23).

Measures:

“For the purpose of the present study, only children were included who had valid data on 4 weekdays and 1 weekend day.” I want to ask about valid data condition.

Response: We have chosen an approach that is well accepted in the international scientific literature. However, although seen as a “gold standard” for physical activity assessment in epidemiological research, accelerometry is not without disadvantages and also requires subjective decisions by the researchers. The fact that many children do not wear the device across the entire day or forget to put on the device on some days can certainly be seen as an issue that influences on the precision of the estimates. This is why researchers generally set minimal standards for wear time. However, accelerometer wear time can vary considerably even between children who fulfill minimal requirements. To deal with this issue, we controlled all regression analyses for accelerometer wear time to ensure that relationships are indeed attributable to differences in MVPA and not a mere reflection of different accelerometer wear times (see revised manuscript, page 12, 12-18). In light of Reviewer #3’s feedback, we have decided to address this issue more frankly in the limitations section.

Does this mean that childrenʹs lifestyle are wrong between weekday and weekend? Is it based on studies shown before (some evidence) or your opinion? Or, (based on) adultsʹ valid data condition? (Page 4, Line 152-155)

Response: The fact that sedentary behaviour and physical activity can vary between school days (during which all children are forced to sit still several hours per day) and weekend days (where children’s physical activity depends more on their individual preferences) is well know from the literature. To be considered as a valid estimate of a child’s overall physical activity behaviour, we have therefore decided that at least one weekend day and four schooldays (with a minimal wear time of 8 hours per day) are required. As mentioned in our manuscript, this procedure has been recommended previously in the literature. In our revised manuscript, we have added the references in which these recommendations can be found (see revised manuscript, page 4, lines 17-18 and page 12, lines 7-10).

Discussion:

If you want to mention the human race, you should represent the percentage of this (white %, black African %, etc.) in your study. If you canʹt describe this, you had better not discuss it more (below sentences are too much). I think it’s a kind of study limitation (lacking this information) (Page 10, Line 296-297).

Response: We do not see this as a limitation. In the present study, we decided against considering the ethnicity of the children as a potential confounding factor or moderator variable. In the South African context, where people have been discriminated for decades on the basis of their ethnic background, using ethnicity as a confounding/moderating factor is a highly sensitive issue and should thus be avoided. In the present paper (see revised manuscript, page 5, lines 8-9), we have decided to mention at the beginning of the results section that most of the children self-identified as black African (51.8%) or coloured (46.7%). This is useful information to characterize our sample. Yet, including this variable as a confounder or moderator must be avoided.

Conclusions:

“Given that school is the place where children spend a considerable proportion of their waking hours, schools have the potential to play a crucial role in the promotion of physical activity and the adoption of a healthy diet.”

From data of device‐based assessments, you cannot judge how long does it spend in school or home and elsewhere. Where MVPA and sedentary activity spent in? It has some possibility of there is room for spending time in their home. This is an overstatement (Page 11, Lines 390-394).

Response: Reviewer #3 raises an important issue; yet, we do not make such claims. Given that children spend a considerable amount of time at school, we believe that our statement – that schools can play an important role in the promotion of physical activity and the adoption of a healthy diet – is accurate. For instance, in line with our statement, data collected in Switzerland show that primary schoolchildren were significantly more active on days with than without physical education lessons (Meyer et al., 2013). [Meyer, U., Roth, R., Zahner, L., Gerber, M., Puder, J., Hebestreit, H., & Kriemler, S. (2013). Contribution of physical education to overall physical activity. Scandinavian Journal of Medicine & Science in Sports, 23, 600-606].

Port Elizabeth region (2. Materials and Methods, 2.2. Setting): You had better add the word “in South Africa”

Response: In line with Reviewer#3’s suggestion, we have added the country (South Africa). However, since South Africa is a large country, we also indicated more precisely the region, in which the study took place. Please note that Port Elizabeth recently changed name and is now called “Gqeberha” (see revised manuscript, page 3, lines 29-30).

ANCOVAs(4. Statistical Analyses): You had better describe the full spell.

Response: We now spelled out ANCOVA upon first mention (see revised manuscript, page 4, line 49) and, subsequently, use the abbreviation (see revised manuscript, page 5, line 1).

Sedentary activity: Sedentary behavior is more proper than sedentary activities.

Response: We fully agree, and hence, we have changed this across the entire manuscript and in Figure 1.

Round 2

Reviewer 3 Report

Dear Autors,

Thank you for the opportunity  to be a part of such an amazing experience.

You deal with my comments politely and properly.

I have no further comments.

I wish your paper will contribute to studies of this field.

Best,